# Efficient Online Learning of Optimal Rankings: Dimensionality Reduction via Gradient Descent

**Dimitris Fotakis**
National Technical University of Athens
fotakis@cs.ntua.gr

**Thanasis Lianeas**
National Technical University of Athens
lianeas@corelab.ntua.gr

**Georgos Piliouras**
Singapore Univ. of Technology & Design
georgios@sutd.edu.sg

**Stratis Skoulakis**
Singapore Univ. of Technology & Design
sskoul@sutd.edu.sg

## Abstract

We consider a natural model of online preference aggregation, where sets of preferred items $R_1, R_2, \ldots, R_t, \ldots$, along with a demand for $k_t$ items in each $R_t$, appear online. Without prior knowledge of $(R_t, k_t)$, the learner maintains a ranking $\pi_t$ aiming that at least $k_t$ items from $R_t$ appear high in $\pi_t$. This is a fundamental problem in preference aggregation with applications to, e.g., ordering product or news items in web pages based on user scrolling and click patterns.

The widely studied *Generalized Min-Sum-Set-Cover* (GMSSC) problem serves as a formal model for the setting above. GMSSC is NP-hard and the standard application of no-regret online learning algorithms is computationally inefficient, because they operate in the space of rankings. In this work, we show how to achieve low regret for GMSSC in polynomial-time. We employ dimensionality reduction from rankings to the space of doubly stochastic matrices, where we apply Online Gradient Descent. A key step is to show how subgradients can be computed efficiently, by solving the dual of a configuration LP. Using oblivious deterministic and randomized rounding schemes, we map doubly stochastic matrices back to rankings with a small loss in the GMSSC objective.

## 1 Introduction

In applications where items are presented to the users sequentially (e.g., web search, news, online shopping, paper bidding), the item ranking is of paramount importance (see e.g., [38, 12, 14, 43, 7]). More often than not, only the items at the first few slots are immediately visible and the users may need to scroll down, in an attempt to discover items that fit their interests best. If this does not happen soon enough, the users get disappointed and either leave the service (in case of news or online shopping, see e.g., the empirical evidence presented in [9]) or settle on a suboptimal action (in case of paper bidding, see e.g., [8]).

To mitigate such situations and increase user retention, modern online services highly optimize item rankings based on user scrolling and click patterns. Each user $t$ is typically represented by her set of preferred items (or item categories) $R_t$. The goal is to maintain an item ranking $\pi_t$ online such that each new user $t$ finds enough of her favorite items at relatively high positions in $\pi_t$ ("enough" is typically user and application dependent). A typical (but somewhat simplifying) assumption is that the user dis-utility is proportional to how deep in $\pi_t$ the user should reach before that happens.

The widely studied *Generalized Min-Sum Set Cover* (GMSSC) problem (see e.g., [28] for a short survey) provides an elegant formal model for the practical setting above. In (the offline version of)

GMSSC, we are given a set $U = \{1, \ldots, n\}$ of $n$ items and a sequence of requests $R_1, \ldots, R_T \subseteq U$. Each request $R \subseteq U$ is associated with a demand (or covering requirement) $\mathrm{K}(R) \in \{1, \ldots, |R|\}$. The *access cost* of a request $R$ wrt. an item ranking (or permutation) $\pi$ is the index of the $\mathrm{K}(R)$-th element from $R$ in $\pi$. Formally,

$$\mathrm{AccessCost}(\pi, R) = \{\text{the first index up to which } \mathrm{K}(R) \text{ elements of } R \text{ appear in } \pi\}. \qquad (1)$$

The goal is to compute a permutation $\pi^* \in [n!]$ of the items in $U$ with minimum total access cost, i.e., $\pi^* = \arg\min_{\pi \in [n!]} \sum_{t=1}^T \mathrm{AccessCost}(\pi, R_t)$.

Due to its mathematical elegance and its connections to many practical applications, GMSSC and its variants have received significant research attention [20, 5, 4, 29]. The special case where the covering requirement is $\mathrm{K}(R_t) = 1$ for all requests $R_t$ is known as *Min-Sum Set Cover* (MSSC). MSSC is NP-hard, admits a natural greedy 4-approximation algorithm and is inapproximable in polynomial time within any ratio less than 4, unless $\mathrm{P} = \mathrm{NP}$ [13]. Approximation algorithms for GMSSC have been considered in a sequence of papers [6, 36, 30] with the state of the art approximation ratio being 12.5. Closing the approximability gap, between 4 and 12.5, for GMSSC remains an interesting open question.

**Generalized Min-Sum Set Cover and Online Learning.** Virtually all previous work on GMSSC (the recent work of [17] is the only exception) assumes that the algorithm knows the request sequence and the covering requirements well in advance. However, in the practical item ranking setting considered above, one should maintain a high quality ranking online, based on little (if any) information about the favorite items and the demand of new users.

Motivated by that, we study GMSSC as an *online learning* problem [21]. I.e., we consider a *learner* that selects permutations over time (without knowledge of future requests), trying to minimize her total access cost, and an adversary that selects requests $R_1, \ldots, R_T$ and their covering requirements, trying to maximize the learner's total access cost. Specifically, at each round $t \geq 1$,

1. The learner selects a permutation $\pi_t$ over the $n$ items, i.e., $\pi_t \in [n!]$.
2. The adversary selects a request $R_t$ with covering requirement $\mathrm{K}(R_t)$.
3. The learner incurs a cost equal to $\mathrm{AccessCost}(\pi_t, R_t)$.

Based on the past requests $R_1, \ldots, R_{t-1}$ only, an *online learning algorithm* selects (possibly with the use of randomization) a permutation $\pi_t$ trying to achieve a total (expected) access cost as close as possible to the total access cost of the optimal permutation $\pi^*$. If the cost of the online learning algorithm is at most $\alpha$ times the cost of the optimal permutation, the algorithm is $\alpha$-*regret* [21]. If $\alpha = 1$, the algorithm is *no-regret*. In this work, we investigate the following question:

**Question 1.** *Is there an online learning algorithm for* GMSSC *that runs in polynomial time and achieves $\alpha$-regret, for some small constant $\alpha \geq 1$?*

Despite a huge volume of work on efficient online learning algorithms and the rich literature on approximation algorithms for GMSSC, Question 1 remains challenging and wide open. Although the *Multiplicative Weights Update* (MWU) algorithm, developed for the general problem of *Learning from Expert Advice*, achieves no-regret for GMSSC, it does not run in polynomial-time. In fact, MWU treats each permutation as a different expert and maintains a weight vector of size $n!$. Even worse, this is inherent to GMSSC, due to the inapproximability result of [13]. Hence, unless $\mathrm{P} = \mathrm{NP}$, MWU's exponential requirements could not be circumvented by a more clever GMSSC-specific implementation, because any polynomial-time $\alpha$-regret online learning algorithm can be turned into a polynomial-time $\alpha$-approximation algorithm for GMSSC. Moreover, the results of [32] on obtaining computationally efficient $\alpha$-regret online learning algorithms from known polynomial time $\alpha$-approximation algorithms for NP-hard optimization problems do not apply to optimizing non-linear objectives (such as the access cost in GMSSC) over permutations.

**Our Approach and Techniques.** Departing from previous work, which was mostly focused on black-box reductions from polynomial-time algorithms to polynomial-time online learning algorithms, e.g., [33, 32], we carefully exploit the structure of permutations and GMSSC, and present polynomial-time low-regret online learning deterministic and randomized algorithms for GMSSC, based on dimensionality reduction and Online Projected Gradient Descent.

Our approach consists of two major steps. The first step is to provide an efficient no-regret polynomial-time learning algorithm for a relaxation of GMSSC defined on doubly stochastic matrices. To

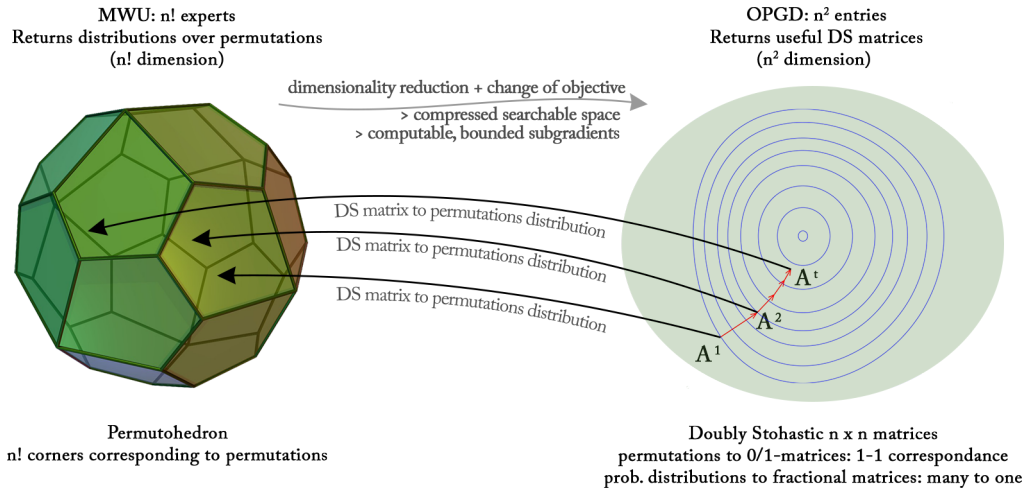

dimensionality reduction + change of objective
> compressed searchable space
> computable, bounded subgradients

DS matrix to permutations distribution
DS matrix to permutations distribution
DS matrix to permutations distribution

$A^t$

$A^2$

$A^1$

Permutohedron
n! corners corresponding to permutations

Doubly Stohastic n x n matrices
permutations to 0/1-matrices: 1-1 correspondance
prob. distributions to fractional matrices: many to one

Figure 1: Our general approach, which is independent of the specific variant of GMSSC.

optimize over doubly stochastic matrices, the learner needs to maintain only $n^2$ values, instead of the $n!$ values required to directly describe distributions over permutations. This *dimensionality reduction* step allows for a polynomial-time no-regret online algorithm for the relaxed version of GMSSC.

The second step is to provide computationally efficient (deterministic and randomized) online rounding schemes that map doubly stochastic matrices back to probability distributions over permutations. The main challenge is to guarantee that the expected access cost of the (possibly random) permutation obtained by rounding is within a factor of $\alpha$ from the access cost of the doubly stochastic matrix representing the solution to the relaxed problem. Once such a bound is established, it directly translates to an $\alpha$-regret online learning algorithm with respect to the optimal permutation for GMSSC. Our approach is summarized in Figure 1.

**Designing and Solving the Relaxed Online Learning Problem.** For the relaxed version of GMSSC, we note that any permutation $\pi$ corresponds to an *integral* doubly stochastic matrix $A^\pi$, with $A^\pi[i,j] = 1$ iff $\pi(j) = i$. Moreover for any request $R$, each doubly stochastic matrix is associated with a *fractional access cost*. For integral doubly stochastic matrices, the fractional access cost is practically identical to the access cost of GMSSC in the respective permutation.

The fractional access cost is given by the optimal solution of an (exponentially large) configuration linear program (LP) that relaxes GMSSC to doubly stochastic matrices (see also [30]), and is a convex function. Thus, we can use *Online Projected Gradient Descent* (OPGD) [44] to produce a *no-regret* sequence of doubly stochastic matrices for the GMSSC relaxation. However, the efficient computation of the subgradient is far from trivial, due to the exponential size of the configuration LP. A key technical step is to show that the subgradient of the configuration LP can be computed in polynomial time, by solving its dual (which is of exponential size, so we resort to the elipsoid method and use an appropriate separation oracle).

**Our Results.** In nutshell, we resolve Question 1 in the affirmative. In addition to solving the relaxed version of GMSSC by a polynomial-time no-regret online learning algorithm, as described above, we present a *polynomial-time randomized rounding scheme* that maps any doubly stochastic matrix to a probability distribution on permutations. The expected access cost of such a probability distribution is at most 28 times the fractional access cost of the corresponding doubly stochastic matrix. Consequently, a 28-regret *polynomial-time randomized online learning algorithm* for GMSSC can be derived by applying, in each round, this rounding scheme to the doubly stochastic matrix $A^t$, produced by OPGD. For the important special case of MSSC, we improve the regret bound to 11.713 via a similar randomized rounding scheme that exploits the fact that $K(R) = 1$ for all requests.

We also present a *polynomial-time deterministic rounding scheme* mapping any (possibly fractional) doubly stochastic matrix to permutations. As before, applying this scheme to the sequence of doubly stochastic matrices produced by OPGD for the relaxation of GMSSC leads to a *polynomial-time*

*deterministic online learning algorithm* with regret $2\max_t |R_t|$ for MSSC. Such a nontrivial upper bound on the regret of deterministic online learning algorithms is rather surprising. Typically, learners that select their actions deterministically fail to achieve any nontrivial regret bounds (e.g., recall that in Learning From Expert Advice, any deterministic online algorithm has $\Omega(\#\text{experts})$ regret, which in case of MSSC is $n!$). Although $2\max_t |R_t|$ is not constant, one should expect that the requests are rather small in most practical applications. The above result is approximately tight, since any deterministic online learning algorithm must have regret at least $\max_t |R_t|/2$ [17, Theorem 1.1]. We should also highlight that the positive results of [17] do not imply the existence of computationally efficient online learning algorithms for MSSC, because their approach is based on the MWU algorithm and uses a state space of $n!$. The state of the art and our results (in bold) are summarized below.

|  | Running Time | Upper Bound (Regret) | Lower Bound (Regret) |
| --- | --- | --- | --- |
| *GMSSC* | Exponential (MWU) | 1 | 1 |
| *GMSSC* | Polynomial | **28** | 4 (any polynomial time) |
| *MSSC* | Polynomial | **11.713** | 4 (any polynomial time) |
| *MSSC* | Exponential (deterministic) | $2 \cdot \max_t |R_t|$ | $\frac{\max_t |R_t|}{2}$ (any deterministic) |
| *MSSC* | Polynomial (deterministic) | $\mathbf{2 \cdot \max_t |R_t|}$ | $\frac{\max_t |R_t|}{2}$ (any deterministic) |

**Related Work.** Our work relates with the long line of research concerning the design of time-efficient online learning algorithms in various combinatorial domains in which the number of possible actions is exponentially large. Such domains include online routing [26, 3], selection of permutations [40, 42, 2, 27], selection of binary search trees [41], submodular minimization/maximization [23, 31, 37], matrix completion [24], contextual bandits [1, 11] and many more.

Apart from the above line of works, concerning the design of time-efficient online learning algorithms in specific settings, another line of research studies the design of online learning algorithms considering *black-box access* in offline algorithms [33, 35, 39, 34, 15, 10, 25, 32, 18, 19, 22]. In their seminal work [33], Kalai et al. showed how a polynomial-time algorithm solving optimally the underlying combinatorial problem, can be converted into a no-regret polynomial-time online learning algorithm. The result of Kalai et al. was subsequently improved [35, 39, 34] for settings in which the underlying problem can be (optimally) solved by a specific approach, such as dynamic programming. Although there do not exist such general reductions for $\alpha$-approximation (offline) algorithms (without taking into account the combinatorial structure of each specific setting [25]), Kakade et al. presented such a reduction for the (fairly general) class of *linear optimization problems* [32]. Their result was subsequently improved by [18, 19, 22]. We remark that the above results do not apply in our setting since GMSSC can neither be optimally solved in polynomial-time nor is a *linear optimization problem*.

Finally our works also relates with a recent line of research studying time-efficient online learning algorithms in settings related to selection of permutations and rankings [42, 2, 27, 38, 43]. The setting considered in [42, 2, 27] is very similar to GMSSC with the difference that once request $R_t$ is revealed, the learner pays the sum of the positions of $R_t$'s elements in permutation $\pi_t$. In this case the underlying combinatorial optimization problem can be solved in polynomial-time meaning that the reduction of [33] produces a time-efficient no-regret online learning algorithm. As a result, all the above works focus on improving the vanishing rate of time-average regret. The setting considered in [38] is based on the *submodular maximization problem*. In particular, the number of available positions is less than the number of elements, while the cost of the selected assignment depends on the set of elements assigned to the slots (their order does not matter). Although this problem is NP-hard, it admits an $(1 - 1/e)$-approximation algorithm which is matched by the presented online learning algorithm. Finally in [43], the cost of the selected permutation is its distance from a permutation selected by the adversary. In this case the underlying combinatorial optimization problem admits an offline $11/9$-approximation algorithm, while a polynomial-time online learning algorithm with $3/2$-regret is presented. We note that GMSSC admits a fairly more complicated combinatorial structure from the above settings and this is indicated by its 4 inapproximability result.

## 2 Definitions and Notation

**Definition 1** (Subgradient). *Given a function $f : D \mapsto \mathbb{R}$, with $D \subseteq \mathbb{R}^n$, a vector $g \in \mathbb{R}^n$ is a subgradient of $f$ at point $x \in \mathbb{R}^n$, denoted $g \in \partial F(x)$, if $f(y) \geq f(x) + g^\top (y - x)$, for all $y \in D$.*

A matrix $A \in [0,1]^{n \times n}$ is *doubly stochastic*, if (i) $A_{ij} \geq 0$, for all $1 \leq i, j \leq n$, (ii) $\sum_{i=1}^n A_{ij} = 1$, for all $1 \leq j \leq n$, and (iii) $\sum_{j=1}^n A_{ij} = 1$, for all $1 \leq i \leq n$. We let DS denote the set of $n \times n$ doubly stochastic matrices.

Any permutation $\pi \in [n!]$ can be represented by an *integral* doubly-stochastic $A^\pi$, where $A_{ij}^\pi = 1$ iff $\pi(j) = i$. Under this representation, the access cost of GMSSC, defined in (1), becomes:

$$\text{AccessCost}(\pi, R) = \sum_{i=1}^n \min \left\{ 1, \left( \text{K(R)} - \sum_{j=1}^{i-1} \sum_{e \in R} A_{ej}^\pi \right)_+ \right\}, \tag{2}$$

where we define $(x - y)_+ = \max\{x - y, 0\}$.

A key notion for our algorithms and analysis is that of *configurations*. Given a request $R \subset U$, a *configuration* $F$ is an assignment of the elements $e \in R$ to positions $j \in [n]$ such that no two elements $e, e' \in R$ share the same position. Intuitively, a configuration wrt. a request $R$ is the set of all permutations $\pi \in [n!]$ with the elements of $R$ in the exact same positions as indicated by $F$. As a result, all permutations $\pi \in [n!]$ that agree with a configuration $F$ wrt. a request $R$ have the same $\text{AccessCost}(\pi, R)$. In the following, $\text{F}(R)$ denotes the set of all configurations wrt. a request $R$ and $C_F$ denotes the access cost $\text{AccessCost}(\pi, R)$ of any permutation $\pi \in [n!]$ that agrees with the configuration $F \in \text{F}(R)$.

**Example 1.** *Let $R = \{2, 5, 7\}$ with $\text{K}(R) = 2$. The configuration $F_1 = \{(2, 3), (5, 1), (7, 10)\}$ stands for the set of permutations $\pi \in [n!]$ in which (i) $\pi(3) = 2$, (ii) $\pi(1) = 5$, and (iii) $\pi(10) = 7$. The configuration $F_1$ is valid (i.e., $F_1 \in \text{F}(R)$), because no elements of $R$ share the same position. Moreover, $C_{F_1} = 3$, because any permutation $\pi$ agreeing with $F$ has cost $3$ for $\text{K}(R) = 2$. Similarly, for the configuration $F_2 = \{(2, 3), (5, 1), (7, 2)\}$, $C_{F_2} = 2$.*

## 3 Solving a Relaxation of Generalized Min-Sum Set Cover

Next, we present an online learning problem for a relaxed version of GMSSC in the space of doubly stochastic matrices. Specifically, we consider an online learning setting where, in each round $t \geq 1$,

1. The learner selects a doubly stochastic matrix $A^t \in \text{DS}$.
2. The adversary selects a request $R_t$ with covering requirements $\text{K}(R_t)$.
3. The learner incurs the *fractional access cost* $\text{FAC}_{R_t}(A^t)$ presented in Definition 2.

**Definition 2** (Fractional Access Cost). *Given a request $R$ with covering requirements $\text{K}(R)$, the fractional access cost of a doubly stochastic matrix $A$, denoted as $\text{FAC}_R(A)$ is the value of the following linear program:*

$$\begin{aligned}
\text{minimize} \quad & \sum_{F \in \text{F}(R)} C_F \cdot y_F + \frac{n^4}{\epsilon} \cdot \sum_{e \in R} \sum_{j=1}^n \left| A_{ej} - \sum_{F:(e,j) \in F} y_F \right| \\
\text{subject to} \quad & \sum_{\text{F} \in \text{F}(R)} y_F = 1 \\
& y_F \geq 0, \ \forall F \in \text{F}(R)
\end{aligned} \tag{FLP}$$

We always assume a fixed accuracy parameter $\epsilon$ (see also Theorem 1 about the role of $\epsilon$). Hence, for simplicity, we always ignore the dependence of $\text{FAC}_R(A)$ on $\epsilon$. We should highlight that we need to deviate from the configuration LP of [30, Sec. 2], because *OPGD* requires an upper bound in the subgradient's norm. The $n^4$ term in (FLP) was appropriately selected so as to ensure that the *access cost* of the probability distribution on permutations produced by a doubly stochastic matrix is upper bounded by its *fractional access cost* (see Section B.1).

An important property of the fractional access cost in Definition 2 is that for all integral doubly stochastic matrices, it is bounded from above by the access cost of GMSSC in (2). For that, simply

note that a feasible solution is setting $y_F = 1$ only for the configuration that "agrees" in the resources of $R$ with the permutation of the integral matrix $A$.

**Corollary 1.** *For any integral doubly stochastic matrix $A^\pi$ corresponding to a permutation $\pi \in [n!]$,*

$$\text{FAC}_R(A^\pi) \leq \text{AccessCost}(\pi, R).$$

For $A^1, A^2 \in \text{DS}$, it is $\text{FAC}_{R_t}\left(\lambda A^1 + (1 - \lambda)A^2\right) \leq \lambda \cdot \text{FAC}_{R_t}\left(A^1\right) + (1 - \lambda) \cdot \text{FAC}_{R_t}\left(A^2\right)$, meaning that $\text{FAC}_{R_t}(\cdot)$ is a convex function in the space of doubly stochastic matrices. Since doubly stochastic matrices form a convex set, *Online Projected Gradient Descent* [44] is a *no-regret* online learning algorithm for the relaxed version of GMSSC.

### 3.1 Implementing Online Gradient Descent in Polynomial-time

*Online Gradient Descent* requires, in each round $t$, the computation of a subgradient of the fractional access cost $\text{FAC}_{R_t}(A_t)$ (see also Definition 1). Specifically, given a request $R$ and a doubly stochastic matrix $A$, a vector $g \in \mathbb{R}^{n^2}$ belongs to the subgradient $\partial\text{FAC}_R(A)$, if for any $B \in \text{DS}$,

$$\text{FAC}_R(B) \geq \text{FAC}_R(A) + g^\top (B - A),  \tag{3}$$

where we slightly abuse the notation and think of matrices $A$ and $B$ as vectors in $[0, 1]^{n^2}$.

Computing a subgradient $g \in \partial\text{FAC}_R(A)$ in polynomial-time is far from trivial, because the fractional access cost $\text{FAC}_R(A)$ does not admit a closed form, since its value is determined by the optimal solution to (FLP). Moroever, (FLP) has exponentially many variables $y_F$, one for each configuration $F \in \text{F}(R)$. We next show how to compute a subgradient $g \in \partial\text{FAC}_R(A)$ by using linear programming duality and solving the *dual* of (FLP), which is presented below:

$$\begin{aligned}
\text{maximize} \quad & \lambda + \sum_{e \in R} \sum_{j=1}^{n} A_{ej} \cdot \lambda_{ej} \\
\text{subject to} \quad & \lambda + \sum_{(e,j) \in F} \lambda_{ej} \leq C_F, \text{ for all } F \in \text{F}(R) \\
& |\lambda_{ej}| \leq n^4/\epsilon
\end{aligned} \tag{4}$$

**Lemma 1.** *For any request $R$ and any stochastic matrix $A \in \text{DS}$, let $g \in \mathbb{R}^{n^2}$ denote the vector consisting of the $n^2$ values of the variables $\lambda_{ej}^*$ in the optimal solution of (4). Then, for any $B \in \text{DS}$,*

$$\text{FAC}_R(B) \geq \text{FAC}_R(A) + g^\top (B - A)$$

*Moreover the Euclidean norm of $g$ is upper bounded by $n^5/\epsilon$, i.e., $\|g\|_2 \leq n^5/\epsilon$.*

Lemma 1 shows that a subgradient $g \in \partial\text{FAC}_R(A)$ can be obtained from the solution to the dual LP (4). Although (4) has exponentially many constraints, we can solve it in polynomial-time by the ellipsoid method, through the use of an appropriate separation oracle.[1] In fact, our separation oracle results from a simple modification of the separation oracle in [30, Sec. 2.3] (see also Section A). Now, the reasons for the particular form of fractional access cost in Definition 2 become clear: (i) it allows for efficient computation of the subgradients, and (ii) the dual constraints $|\lambda_{ej}| \leq n^4/\epsilon$ imply that the subgradient's norm is always bounded by $n^5/\epsilon$.

**Remark 1.** *For the Min-Sum Set Cover problem, the use of the ellipsoid method (for the computation of the subgradient vector) can be replaced by a more efficient quadratic-time algorithm (see Appendix B.2).*

Having established polynomial-time computation for the subgradients, Online Projected Gradient Descent takes the form of Algorithm 1 in our specific setting.

**Algorithm 1** Online Projected Gradient Decent in Doubly Stochastic Matrices

---

1: Initially, the player selects the matrix $A^1 = 1/n \cdot 1_{n \times n}$.
2: **for all** rounds $t = 1 \cdots T$ **do**
3:     The adversary selects a request $R_t \subseteq U$ with covering requirements $\mathrm{K}(R_t)$.
4:     The learner receives cost, $\mathrm{FAC}_{R_t}(A^t)$.
5:     The learner computes a subgradient $g_t \in \partial\mathrm{FAC}_{R_t}(A^t)$ by solving the dual of (FLP).
6:     The learner computes the matrix, $\hat{A} = A^t - 2\epsilon \cdot g_t/(n^{4.5}\sqrt{t})$.
7:     The learner adopts the matrix, $A^{t+1} = \arg\min_{A \in \mathrm{DS}} \|A - \hat{A}\|_{\mathrm{F}}$
8: **end for**

---

Step 6 of Algorithm 1 is the *gradient step*. In Online Projected Gradient Descent, this step is performed with step-size $D/(G\sqrt{t})$, where $D$ and $G$ are upper bounds on the diameter of the action space and on the Euclidean norm of the subgradients. In our case, the action space is the set of doubly stochastic matrices. Since $\max_{A,B \in \mathrm{DS}} \|A - B\|_{\mathrm{F}} \leq 2\sqrt{n}$ the parameter $D = 2\sqrt{n}$, and $G = n^5/\epsilon$, by Lemma 1. Hence, our step-size is $2\epsilon/(n^{4.5}\sqrt{t})$. The projection step (Step 7) is implemented in polynomial-time, because projecting to doubly stochastic matrices is a convex problem [16]. We conclude the section by plugging in the parameters $G = n^5/\epsilon$ and $D = 2\sqrt{n}$ to the regret bounds of Online Projected Gradient Descent [44], thus obtaining Theorem 1.

**Theorem 1.** *For any $\epsilon > 0$ and any request sequence $R_1, \ldots, R_T$, the sequence of doubly stochastic matrices $A^1, \ldots, A^T$ produced by Online Projected Gradient Descent (Algorithm 1) satisfies,*
$$\frac{1}{T}\sum_{t=1}^{T} \mathrm{FAC}_{R_t}(A^t) \leq \frac{1}{T}\min_{A \in \mathrm{DS}}\sum_{t=1}^{T} \mathrm{FAC}_{R_t}(A) + O\left(\frac{n^{5.5}}{\epsilon\sqrt{T}}\right).$$

## 4 Converting Doubly Stochastic Matrices to Distributions on Permutations

Next, we present polynomial-time rounding schemes that map a doubly stochastic matrix back to a probability distribution on permutations. Our schemes ensure that the resulting permutation (random or deterministic) has access cost at most $\alpha$ times the fractional access cost of the corresponding doubly stochastic matrix. Combining such schemes with Algorithm 1, we obtain polynomial-time $\alpha$-regret online learning algorithms for GMSSC.

Due to lack of space, we only present the deterministic rounding scheme, which is intuitive and easy to explain. Most of its analysis and the description of the randomized rounding schemes are deferred to the supplementary material.

---

**Algorithm 2** Converting Doubly Stochastic Matrices to Permutations

---

**Input:** A doubly stochastic matrix $A \in \mathrm{DS}$, a parameter $r$ and a parameter $\alpha > 0$.
**Output:** A deterministic permutation $\pi_A \in [n!]$.

1: $\mathrm{Rem} \leftarrow \{1, \ldots, n\}$
2: **for** $k = 1$ to $\lfloor n/r \rfloor$ **do**
3:     Let $R_k$ be any $(1 + \alpha)$-approximate solution to the following problem:

$$\min_{R \subseteq \mathrm{Rem}:|R|=r} \sum_{i=1}^{n} \left(1 - \sum_{j=1}^{i-1}\sum_{e \in R} A_{ej}\right)_+$$

4:     Assign the elements of $R_k$ to positions $(k-1) \cdot r + 1, \ldots, k \cdot r$ of $\pi_A$ in any order.
5:     $\mathrm{Rem} \leftarrow \mathrm{Rem} \setminus R_k$
6: **end for**
7: **return** the resulting permutation $\pi_A \in [n!]$.

---

Algorithm 2 aims to produce a permutation $\pi_A \in [n!]$ from the doubly stochastic matrix $A$ such that the $\mathrm{AccessCost}(\pi_A, R)$ is approximately bounded by $\mathrm{FAR}_R(A)$ for any request $R$ with $|R| \leq r$ and $\mathrm{K}(R) = 1$. Algorithm 2 is based on the following intuitive greedy criterion:

Assign to the first $r$ available positions of $\pi_A$ the elements of the request of size $r$ with minimum fractional cost of Definition 2 wrt. the doubly stochastic matrix $A$. Then, remove these elements and repeat.

Unfortunately the greedy step above involves the solution to an NP-hard optimization problem. Nevertheless, we can approximate it with an FPTAS (Fully Polynomial-Time Approximation Scheme). The $(1+\alpha)$-approximation algorithm used in Step 3 of Algorithm 2 runs in $\Theta(n^4 r^3/\alpha^2)$ and is presented and analyzed in Section B.5. Theorem 2 (proved in Section B.3) summarizes the guarantees on the access cost of a permutation $\pi_A$ produced by Algorithm 2.

**Theorem 2.** *Let $\pi_A$ denote the permutation produced by Algorithm 2 when the doubly stochastic matrix $A$ is given as input. Then for any request $R$ with $\mathrm{K}(R) = 1$ and $|R| \leq r$,*

$$\mathrm{AccessCost}(\pi_A, R) \leq 2(1+\epsilon)(1+\alpha)^2 r \cdot \mathrm{FAC}_R(A),$$

*with $\epsilon > 0$ as in Definition 2. Moreover, Step 3, can be implemented in $\Theta(n^4 r^3/\alpha^2)$ steps.*

We now show how Algorithm 1 and Algorithm 2 can be combined to produce a *polynomial-time deterministic online learning algorithm* for MSSC with regret roughly $2 \max_{1 \leq t \leq T} |R_t|$. For any adversarially selected sequence of requests $R_1, \ldots, R_T$ with $\mathrm{K}(R_t) = 1$ and $|R_t| \leq r$, the learner runs Algorithm 1 *in the background*, while at each round $t$ uses Algorithm 2 to produce the permutation $\pi_{A^t}$ by the doubly stochastic matrix $A^t \in \mathrm{DS}$. Then,

$$
\begin{aligned}
\frac{1}{T} \sum_{t=1}^{T} \mathrm{AccessCost}(\pi_{A^t}, R_t) & \leq \frac{1}{T} \cdot \sum_{t=1}^{T} 2(1+\epsilon)(1+\alpha)^2 r \cdot \mathrm{FAC}_{R_t}(A^t) \\
& \leq \frac{2r}{T}(1+\epsilon)(1+\alpha)^2 \cdot \min_{A \in \mathrm{DS}} \sum_{t=1}^{T} \mathrm{FAC}_{R^t}(A) + O\left(\frac{n^{5.5}}{\epsilon\sqrt{T}}\right) \\
& \leq \frac{2r}{T}(1+\epsilon)(1+\alpha)^2 \cdot \sum_{t=1}^{T} \mathrm{FAC}_{R^t}(A^{\pi^*}) + O\left(\frac{n^{5.5}}{\epsilon\sqrt{T}}\right) \\
& \leq \frac{2r}{T}(1+\epsilon)(1+\alpha)^2 \cdot \sum_{t=1}^{T} \mathrm{AccessCost}(\pi^*, R^t) + O\left(\frac{n^{5.5}}{\epsilon\sqrt{T}}\right)
\end{aligned}
$$

The first inequality follows by Theorem 2, the second by Theorem 1 and the last by Corollary 1.

Via the use of *randomized rounding schemes* we can substantially improve both on the assumptions and the guarantee of Theorem 2. Algorithm 3 (presented in Section B.1), describes such a scheme that converts any doubly stochastic matrix $A$ to a probability distribution over permutations, while Theorem 3 (also proven in Section B.1) establishes an approximation guarantee (arbitrarily) close to 28 on the expected access cost.

**Theorem 3.** *Let $\mathrm{P_A}$ denote the probability distribution over permutations that Algorithm 3 produces given as input an $A \in \mathrm{DS}$. For any request $R$,*

$$\mathbb{E}_{\pi \sim \mathrm{P_A}} [\mathrm{AccessCost}(\pi, R)] \leq 28(1+\epsilon) \cdot \mathrm{FAC}_R(A)$$

*where $\epsilon > 0$ is the parameter used in Definition 2.*

Using Theorem 3 instead of Theorem 2 in the previously exhibited analysis, implies that combining Algorithms 1 and 3 leads to a *polynomial-time randomized online learning algorithm* for GMSSC with $28(1+\epsilon)$ regret.

In Section B.2 we improve Theorem 3 for the the special case of MSSC. The randomized rounding scheme described in Algorithm 4 admits the approximation guarantee of Theorem 4, which implies a *polynomial-time randomized online learning algorithm* for MSSC with $11.713(1+\epsilon)$ regret

**Theorem 4.** *Let $\mathrm{P_A}$ denote the probability distribution over permutations that Algorithm 4 produces given as input an $A \in \mathrm{DS}$. For any request $R$ with covering requirement $\mathrm{K}(R) = 1$,*

$$\mathbb{E}_{\pi \sim \mathrm{P_A}} [\mathrm{AccessCost}(\pi, R)] \leq 11.713(1+\epsilon) \cdot \mathrm{FAC}_R(A)$$

*where $\epsilon > 0$ is the parameter used in Definition 2.*

# 5   Experimental Evaluations

In this section we provide experimental evaluations of all the proposed online learning algorithms (both deterministic and randomized) for *Min-Sum Set Cover*. Surprisingly enough our simulations seem to suggest that the *deterministic rounding scheme* proposed in Algorithm 2, performs significantly better than its theoretical guarantee, stated in Theorem 2, that associates its regret with the cardinality of the sets. The following figures illustrate the performance of Algorithm 2 and Algorithm 4, and compare it with the performance of the offline algorithm proposed by Feige et al. [13] and the performance of selecting a permutation uniformly at random at each round. In the left figure each request contains either element 1 or 2 and four additional randomly selected elements, while in the right figure each request contains one of the elements $\{1, 2, 3, 4, 5\}$ and nine more randomly selected elements.[2] We remark that in our experimental evaluations, we solve the optimization problem of Step 3 in Algorithm 2 through a simple heuristic that we present in Appendix B.6, while for the computation of the subgradients we use the formula presented in Corollary 3. The code used for the presented simulations can be found at `https://github.com/sskoul/ID2216`.

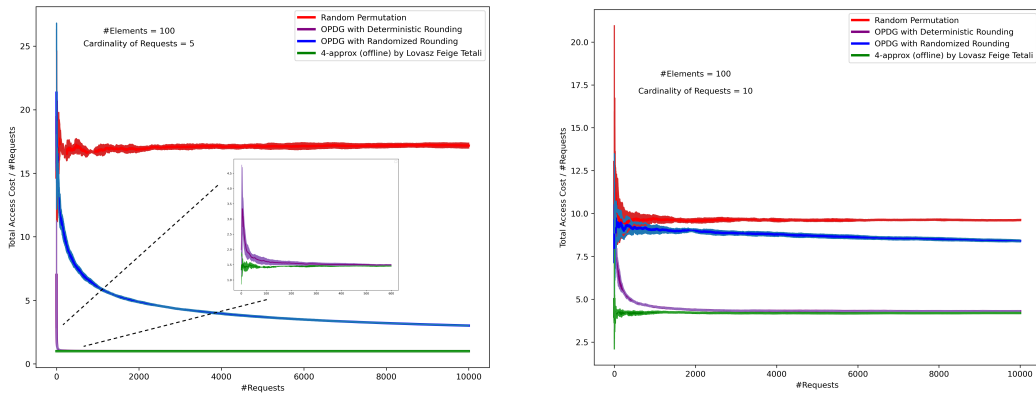

# 6   Conclusion

This work examines polynomial-time online learning algorithms for *(Generalized) Min-Sum Set Cover*. Our results are based on solving a relaxed online learning problem of smaller dimension via *Online Projected Gradient Descent*, the solution of which is transformed at each round into a solution of the initial action space with bounded increase in the cost. To do so, the cost function of the relaxed online learning problem is defined by the value of a linear program with exponentially many constraints. Despite its exponential size, we show that the subgradients can be efficiently computed via associating them with the variables of the LP' s *dual*. We believe that the bridge between online learning algorithms (e.g. online projected gradient descent) and traditional algorithmic tools (e.g. duality, separation oracles, deterministic/randomized rounding schemes), introduced in this work, is a promising new framework for the design of efficient online learning algorithms in high dimensional combinatorial domains. Finally closing the gap between our regret bounds and the lower bound of 4, which holds for polynomial-time online learning algorithms for MSSC, is an interesting open problem.

## Broader Impact

We are living in a world of abundance, where each individual is provided myriad of options in terms of available products and services (e.g. music selection, movies etc.). Unfortunately this overabundance makes the cost of exploring all of them prohibitively large. This problem is only compounded by the fast turn around of new trends at a seemingly ever increasing rate. Our algorithmic techniques provide a practically applicable methodology for managing this complexity.

## Funding Disclosure

Dimitris Fotakis and Thanasis Lianeas are supported by the Hellenic Foundation for Research and Innovation (H.F.R.I.) under the "First Call for H.F.R.I. Research Projects to support Faculty members and Researchers' and the procurement of high-cost research equipment grant", project BALSAM, HFRI-FM17-1424. Stratis Skoulakis was supported by NRF 2018 Fellowship NRF-NRFF2018-07. G. Piliouras gratefully acknowledges AcRF Tier-2 grant (Ministry of Education – Singapore) 2016-T2-1-170, grant PIE-SGP-AI-2018-01, NRF2019-NRF-ANR095 ALIAS grant and NRF 2018 Fellowship NRF-NRFF2018-07 (National Research Foundation Singapore).

## Footnotes

[1]Interestingly, GMSSC seems to be the most general version of min-sum-set-cover-like ranking problems that allow for an efficient subgradient computation through the dual of the configuration LP (FLP). E.g., for the version of Min-Sum-Set-Cover with submodular costs considered in [4], determining the feasibility of a potential solution to (4) is NP-hard. This is true even for very special case where the cover time function used in [4] is additive.

[2]In the subsequent figures the curves describing the performance of each algorithm are placed in the following top-down order i) *Selecting a permutation uniformly at random*, ii) *Algorithm 2*, iii) *Algorithm 4* and iv) Feige-Lovasz-Tetali algorithm [13].

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
