[Supplementary Material]

# Supplementary Material
# Efficient Online Learning of Optimal Rankings: Dimensionality Reduction via Gradient Descent

## A  Omitted Proofs of Section 3

*Proof of Lemma 1.* To simplify notation, let $\lambda^*(A), \lambda^*_{ej}(A)$ denote the values of the variables $\lambda^*(A), \lambda^*(A)_{ej}$ in the optimal solution of the dual program written with respect to doubly stochastic matrix $A \in \mathrm{DS}$. Respectively $\lambda^*(B), \lambda^*_{ej}(B)$ for the doubly stochastic matrix $B \in \mathrm{DS}$. By strong duality, we have that

$$\mathrm{FAC}_R(A) = \lambda^*(A) + \sum_{e \in R} \sum_{j=1}^n A_{ej} \cdot \lambda^*_{ej}(A) \text{ and } \mathrm{FAC}_R(B) = \lambda^*(B) + \sum_{e \in R} \sum_{j=1}^n B_{ej} \cdot \lambda^*_{ej}(B)$$

Since matrices $A$ and $B$ only affect the objective function of the dual and not its constraints, the solution $\lambda^*(A), \lambda^*_{ej}(A)$ is a feasible solution for the dual program written according to matrix $B$. By the optimality of $\lambda^*(B), \lambda^*_{ej}(B)$ we get,

$$\mathrm{FAC}_R(B) = \lambda^*(B) + \sum_{e \in R} \sum_{j=1}^n B_{ej} \cdot \lambda^*_{ej}(B) \geq \lambda^*(A) + \sum_{e \in R} \sum_{j=1}^n B_{ej} \cdot \lambda^*_{ej}(A)$$

As a result, we get that $\mathrm{FAC}_R(B) - \mathrm{FAC}_R(A) \geq \sum_{e \in R} \sum_{j=1}^n \lambda^*_{ej}(A) \cdot (B_{ej} - A_{ej})$ implying that the vector $g$ containing the $\lambda^*_{ej}(A)$'s, is a subgradient of $\mathrm{FAC}_R(\cdot)$ at point $A$, i.e., $g \in \partial \mathrm{FAC}_R(A)$. The inequality $\|g\|_2 \leq n^5/\epsilon$ directly follows by the fact that $|\lambda^*(A)_{ej}| \leq n^4/\epsilon$. $\qquad\square$

*Separation Oracle for the LP in Equation 4:* The dual linear program of (4) is differs from the $\mathrm{LP}_{\mathrm{dual}}$ in [30, Sec. 2.2] only in the constraints $|\lambda_{ej}| \leq n^4/\epsilon$, which are only present in (4). [30, Sec. 2.2] present a separation oracle for their $\mathrm{LP}_{\mathrm{dual}}$ (i.e., for (4), without the constraints $|\lambda_{ej}| \leq n^4/\epsilon$), which is based on formulating and solving a min-cost flow problem. Since, in case of (4), the we have only $n^2$ additional constraints $|\lambda_{ej}| \leq n^4/\epsilon$, we can first check whether these constraints are satisfied by the current solution and then run the separation oracle of [30].

## B  Omitted Proofs of Section 4

### B.1  Proof of Theorem 3

In Algorithm 3, we present the *online randomized rounding scheme* that combined with *Projected Gradient Descent* (Algorithm 1) produces a *polynomial-time randomized online learning algorithm* for GMSSC with (roughly) 28 regret. The randomized rounding scheme described in Algorithm 3 was introduced by [36] to provide a 28-approximation algorithm for the (offline) GMSSC. [36] proved that this randomized rounding scheme produces a random permutation with access cost at most 28 times greater than the optimal fractional value of the LP relaxation of GMSSC introduced in [6]. We remark that this LP relaxation cannot be *translated* to an equivalent *relaxed online learning problem* as the one we formulated using the *fractional access cost* of Definition 2. The goal of the section is to prove Theorem 3 which extends the result of [36] to the *fractional access cost* of Definition 2.

---
**Algorithm 3** Converting Doubly Stochastic Matrices to Probability Distributions over Permutations
---
**Input:** A doubly stochastic matrix $A \in \mathrm{DS}$.
**Output:** A probability distribution over permutations, $\mathrm{P}_A \sim \pi \in [n!]$

1: Randomly pick $\alpha \in (0,1)$ with probability density function $f(\alpha) = 2\alpha$.
2: Set $B \leftarrow (5.03/\alpha) \cdot A$
3: **for** all elements $e = 1$ to $n$ **do**
4:     **for** all positions $j = 1$ to $\lfloor n/2 \rfloor$ **do**
5:         $B_{e,2j} \leftarrow B_{e,2j} + B_{e,j}$.
6:     **end for**
7: **end for**
8: **for** all elements $e = 1$ to $n$ **do**
9:     Pick $\alpha_e$ uniformly at random in $[0,1]$.
10:     Find the effecive index $i_e^\alpha \leftarrow \arg\max_i \{i : \sum_{j=1}^{i-1} B_{ej} < \alpha_e\}$.
11: **end for**
12: Output the elements according to the order of $i_e$'s.

---

**Definition 3.** *For a request $R$ with covering requirements $\mathrm{K}(R)$, we define the cost $\mathrm{SW}_R : \mathrm{DS} \mapsto \mathbb{R}$ on the doubly stochastic matrices as follows: For any doubly stochastic matrix $A \in \mathrm{DS}$, the value $\mathrm{SW}_R(A)$ equals the value of the following linear program,*

$$\text{minimize} \quad \sum_{i=1}^{n} (1 - z_i)$$

$$\text{subject to} \quad (\mathrm{K}(R) - |M|) \cdot z_i \leq \sum_{j=1}^{i-1} \sum_{e \in R \setminus M} A_{ej} \text{ for all } M \subseteq R$$

$$z_i \in [0,1] \text{ for all } 1 \leq i \leq n$$

**Lemma 2.** *[36] For any doubly stochastic matrix $A \in \mathrm{DS}$,*

$$\mathop{\mathbb{E}}_{\pi \sim \mathrm{P}_A} [\mathrm{AccessCost}(\pi, R)] \leq 28 \cdot \mathrm{SW}_R(A)$$

*where $\mathrm{P}_A$ is the probability distribution over the permutation produced by Algorithm 3 when the matrix $A$ was given as input.*

In Lemma 3 we associate the cost $\mathrm{SW}_R(\cdot)$ of Definition 3 with the *fractional access cost* $\mathrm{FAC}_R(\cdot)$ of Definition 2. Then Theorem 3 directly follows by Lemma 2 and Lemma 3.

**Lemma 3.** *For any doubly stochastic matrix $A \in \mathrm{DS}$,*

$$\mathrm{SW}_R(A) \leq (1 + \epsilon) \cdot \mathrm{FAC}_R(A)$$

*where $\epsilon > 0$ is the parameter of the linear program (FLP) in Definition 2.*

*Proof.* Starting from the optimal solution $y_F$ of the linear program (FLP) of $\mathrm{FAC}_R(A)$ in Definition 2, we construct a feasible solution for the linear program of $\mathrm{SW}_R(A)$ of Definition 3 with cost approximately bounded by $(1 + \epsilon) \cdot \mathrm{FAC}_R(A)$. We first prove Claim 1 that is crucial for the subsequent analysis.

**Claim 1.** *For any element $e \in R$ and position $1 \leq j \leq n$, $|A_{ej} - \sum_{F:(e,je,e,j) \in F} y_F^*| \leq \epsilon/n^3$.*

*Proof.* Since $A$ is a doubly stochastic matrix, by the Birkhoff-von Neumann theorem there exists a vector $\hat{y}$ with $\hat{y}_F \geq 0$ and $\sum_{F \in \mathrm{F}(R)} \hat{y}_F = 1$ such that

$$|A_{ej} - \sum_{F:(e,j) \in F} \hat{y}_F| = 0 \text{ for all } e \in R \text{ and } 1 \leq j \leq n$$

Since $y^*$ is the optimal solution, we have that

$$\sum_{F \in \mathrm{F}(R)} C_F \cdot y_F^* + \frac{n^4}{\epsilon} \cdot \sum_{e \in R} \sum_{j=1}^{n} |A_{ej} - \sum_{F:(e,j) \in F} y_F^*| \leq \sum_{F \in \mathrm{F}(R)} C_F \cdot \hat{y}_F.$$

Now the claim follows by the fact that $1 \leq C_F \leq n$, $\hat{y}_F \geq 0$ and $\sum_{F \in \mathrm{F}(R)} \hat{y}_F = 1$. $\qquad\square$

Having established Claim 1, we construct the solution $z^*$ that is feasible for the linear program of Definition 3 and its value (under the linear program of Definition 3), is upper bounded by $(1 + \epsilon) \cdot \mathrm{FAC}_R(A)$. For each position $1 \leq i \leq n$,

$$z_i^* = \left( \sum_{F \in \mathrm{F}(R): C_F \leq i-1} y_F^* - \frac{\epsilon}{n} \right)_+$$

We first prove that $z^*$ is feasible for the linear program of Definition 3. At first observe that in case $z_i^* = 0$ or $\mathrm{K}(R) - |M| \leq 0$ for some $M \subseteq R$, the constraint $(\mathrm{K}(R) - |M|) \cdot z_i \leq \sum_{j=1}^{i-1} \sum_{e \in R \setminus M} A_{ej}$ is trivially satisfied. We thus turn our attention in the cases where $z_i^* = \sum_{F: C_F \leq i-1} y_F^* - \epsilon/n > 0$ and $K(R) - |M| \geq 1$ (recall, $K(R)$ and $|M|$ are integers). Applying Claim 1 we get that,

$$
\begin{aligned}
\sum_{e \in R \setminus M} \sum_{j=1}^{i-1} A_{ej} &\geq \sum_{e \in R \setminus M} \sum_{j=1}^{i-1} \left( \sum_{F:(e,j)\in F} y_F^* - \epsilon/n^3 \right) \\
&\geq \sum_{e \in R \setminus M} \sum_{j=1}^{i-1} \sum_{F:(e,j)\in F} y_F^* - \epsilon/n \\
&= \sum_{F \in \mathrm{F}(R)} y_F^* \sum_{e \in R \setminus M} \sum_{j=1}^{i-1} \mathbf{1}[(e,j) \in F] - \epsilon/n \\
&\geq \sum_{F: C_F < i} y_F^* \sum_{e \in R \setminus M} \sum_{j=1}^{i-1} \mathbf{1}[(e,j) \in F] - \epsilon/n \\
&\geq (K(R) - |M|) \sum_{F: C_F < i} y_F^* - \epsilon/n \\
&= (K(R) - |M|) \cdot z_i^* + \epsilon \frac{K(R) - |M|}{n} - \epsilon/n \\
&\geq (K(R) - |M|) \cdot z_i^*
\end{aligned}
$$

where the second to last inequality follows from $C_F < i$, and the last equation and the last inequality follow from $z_i^* + \epsilon/n = \sum_{F: C_F \leq i-1} y_F^*$ and $\mathrm{K}(R) - |M| \geq 1$, respectively.

We complete the proof of Lemma 3 by showing that $\sum_{i=1}^n (1 - z_i^*) \leq (1 + \epsilon) \cdot \mathrm{FAC}_R(A)$.

$$
\begin{aligned}
\mathrm{SW}_R(A) &\leq \sum_{i=1}^n (1 - z_i^*) \\
&\leq \sum_{i=1}^n \left( 1 - \sum_{F: C_F < i} y_F^* + \epsilon/n \right) \\
&= \sum_{i=1}^n \left( 1 - \sum_{F: C_F < i} y_F^* \right) + \epsilon \\
&= \sum_{i=1}^n \sum_{F: C_F \geq i} y_F^* + \epsilon \\
&= \sum_{F \in \mathrm{F}(R)} C_F \cdot y_F^* + \epsilon \\
&\leq (1 + \epsilon) \cdot \mathrm{FAC}_R(A)
\end{aligned}
$$

$\square$

## B.2  Proof of Theorem 4

We first present the online sampling scheme, described in Algorithm 4, that produces the 11.713 guarantee of Theorem 4.

**Algorithm 4** Converting Doubly Stochastic Matrices to Probability Distribution (the case of MSSC)

**Input:** A doubly stochastic matrix $A \in \mathrm{DS}$.
**Output:** A probability distribution over permutations, $\mathrm{P}_A \sim \pi \in [n!]$.

 1: Randomly pick $\alpha \in (0,1)$ with probability density function $f(\alpha) = 2\alpha$.
 2: Set $B \leftarrow Q \cdot A$ where $Q \leftarrow 1.6783/\alpha$.
 3: **for** all elements $e = 1$ to $n$ **do**
 4:     **for** all positions $j = 1$ to $\lfloor n/2 \rfloor$ **do**
 5:         $B_{e,2j} \leftarrow B_{e,2j} + B_{e,j}$
 6:     **end for**
 7: **end for**
 8: **for** all elements $e = 1$ to $n$ **do**
 9:     Pick $\alpha_e$ uniformly at random in $[0,1]$.
10:     $i_e \leftarrow \max\{i : \sum_{j=1}^{i-1} B_{ej} < \alpha_e\}$
11: **end for**
12: Output the elements according to the order of $i_e$'s.

We dedicate the rest of the section to prove Theorem 4. Notice that Algorithm 4 is identical to Algorithm 3 with a slight difference in Step 2. Taking advantage of $K(R) = 1$, with tailored analysis, we significantly improve to 11.713 the 28 bound of Lemma 2. Once Lemma 4 below is established, Theorem 4 follows by the exact same steps that Theorem 3 follows using Lemma 2. The proof of Lemma 4 is concluded at the end of the section.

**Lemma 4.** *Let* $\mathrm{P}_A$ *denote the probability distribution over permutations produced by Algorithm 4 when matrix $A$ is given as input. For all requests $R$ with $\mathrm{K}(R) = 1$,*

$$\mathop{\mathbb{E}}_{\pi \sim \mathrm{P}_A}[\mathrm{AccessCost}(\pi, R)] \leq 11.713 \cdot \mathrm{SW}_R(A)$$

*where* $\mathrm{SW}_R(\cdot)$ *is the cost of Definition 3.*

In fact $\mathrm{SW}_R(\cdot)$ takes a simpler form.

**Corollary 2.** *For any request $R$ with covering requirement $\mathrm{K}(R) = 1$, the cost $\mathrm{SW}_R(\cdot)$ of Definition 3 takes the following simpler form,*

$$\mathrm{SW}_R(A) = \sum_{i=1}^{n} \left(1 - \sum_{j=1}^{i-1}\sum_{e \in R} A_{ej}\right)_+$$

**Lemma 5.** *[36] For the matrix $B$ constructed at Step 2 of Algorithm 4, the following holds:*

*1.* $\displaystyle\sum_{j=1}^{2^k i} B_{ej} \geq (k+1) \sum_{j=1}^{i} A_{ej}$

*2.* $\displaystyle\sum_{j=1}^{i}\sum_{e=1}^{n} B_{ej} \leq 2Q \cdot i.$

Condition 2 of Lemma 5 allows for a bound on the expected access cost of the probability distribution produced by Algorithm 4 with respect to the indices $i_e$ of Step 10. This is formally stated below.

**Lemma 6.** *Let* $\mathrm{P}_\alpha^A$ *denote the probability distribution produced in Steps $2 - 11$ of Algorithm 4 for a fixed value of $\alpha$. Then for any request $R$ with covering requirements $\mathrm{K}(R) = 1$,*

$$\mathop{\mathbb{E}}_{\pi \sim \mathrm{P}_\alpha^A}[\mathrm{AccessCost}(\pi, R)] \leq 2Q \cdot \mathbb{E}[\min_{e \in R} i_e] + 1,$$

*with $i_e$ as defined in Step 10 of Algorithm 4.*

*Proof.* Let $O_i^R$ denote the set of elements outside $R$ with index value $i_e \leq i$,

$$O_i^R = \{e \notin R : i_e \leq i\}.$$

Notice that Algorithm 4 orders the elements with respect to the values $i_e$ (Step 12). Since the covering requirements of the request $R$ is $\mathrm{K}(R) = 1$,

$$\mathrm{AccessCost}(\pi, R) \leq |O^R_{\min_{e \in R} i_e}| + 1.$$

The latter holds since $R$ is covered at the first index in which one of its elements appears ($\mathrm{K}(R) = 1$). As a result,

$$\mathop{\mathbb{E}}_{\pi \sim P^A_\alpha}[\mathrm{AccessCost}(\pi, R)] \leq \mathbb{E}[|O^R_{\min_{e \in R} i_e}|] + 1 \leq \sum_{e' \notin R} \Pr[i_{e'} \leq \min_{e \in R} i_e] + 1$$

It is not hard to see that,

$$\sum_{e' \notin R} \Pr[i_{e'} \leq \min_{e \in R} i_e] + 1 = \mathbb{E}[\sum_{e' \notin R} \sum_{j=1}^{\min_{e \in R} i_e} B_{e'j}] + 1 \leq 2Q \cdot \mathbb{E}[\min_{e \in R} i_e] + 1$$

where the first equality follows by the fact that, once $B$ is fixed, $\Pr[i_e \leq k] = \sum_{j=1}^{k} B_{ej}$ (Step 10 of Algorithm 4) and the last inequality follows by Case 2 of Lemma 5. □

**Lemma 7.** *Let $i^\alpha_R$ denote the first position at which $\sum_{j=1}^{i^\alpha_R} \sum_{e \in R} A_{ej} \geq \alpha$ then*

$$\int_0^1 i^\alpha_R \, d\alpha \leq \sum_{i=1}^{n} \left( 1 - \sum_{j=1}^{i-1} \sum_{e \in R} A_{ej} \right)_+ = \mathrm{SW}_R(A)$$

*Proof.* In order to prove Lemma 7, let us assume that a random variable $\beta$ is selected according to the uniform probability distribution in $[0, 1]$, i.e., with density function $f(\beta) = 1$. As a result, $\int_0^1 i^\alpha_R \, d\alpha = \int_0^1 i^\beta_R \, d\beta = \mathbb{E}[i^\beta_R] = \sum_{i=1}^{n} \Pr[i^\beta_R \geq i]$. Since $i^\beta_R$ is the first position at which $\sum_{j=1}^{i^\beta_R} \sum_{e \in R} A_{ej} \geq \beta$,

$$\Pr[i^\beta_R \geq i] = \Pr[\beta > \sum_{j=1}^{i-1} \sum_{e \in R} A_{ej}] = \max \left( 1 - \sum_{j=1}^{i-1} \sum_{e \in S} A_{ej}, 0 \right) \leq \sum_{i=1}^{n} \left( 1 - \sum_{j=1}^{i-1} \sum_{e \in R} A_{ej} \right)_+$$

with the second equality following because $\beta$ is selected according to the uniform distribution in $[0, 1]$. □

To this end we have upper bounded the expected access cost of Algorithm 4 by $\mathbb{E}[\min_{e \in R} i_e]$ (Lemma 4) and lower bounded $\mathrm{SW}_R(A)$ by $\int_0^1 i^\alpha_R \, d\alpha$ (Lemma 7). In Lemma 8 we associate these bounds. At this point the role of Condition 1 of Lemma 5 is revealed.

**Lemma 8.** *Let $i^\alpha_R$ denote the first position at which $\sum_{j=1}^{i^\alpha_R} \sum_{e \in R} A_{ej} \geq \alpha$ then*

$$\mathbb{E}[\min_{e \in R} i_e] \leq i^\alpha_R / (1 - 2e^{-\alpha Q}).$$

*Proof.*

$$\begin{aligned}
\Pr[\min_{e \in R} i_e \geq 2^k \cdot i^\alpha_R + 1] &= \Pi_{e \in R} \Pr[i_e \geq 2^k \cdot i^\alpha_R + 1] \\
&= \Pi_{e \in R} \Pr[\alpha_e > \sum_{j=1}^{2^k \cdot i^\alpha_R} B_{ej}] \\
&= \Pi_{e \in R} \left( 1 - \sum_{j=1}^{2^k \cdot i^\alpha_R} B_{ej} \right)_+
\end{aligned}$$

$$\leq \quad e^{-\sum\limits_{e \in R} \sum\limits_{j=1}^{2^k \cdot i_R^\alpha} B_{ej}}$$

$$\leq \quad e^{-(k+1)Q \sum\limits_{e \in S} \sum\limits_{j=1}^{i_R^\alpha} A_{ej}}$$

$$\leq \quad e^{-(k+1)Q\alpha} = p^{k+1}$$

where the second inequality follows by Case 1 of Lemma 5 and the definition (and manipulation) of matrix $B$ inside Algorithm 4.

$$\mathbb{E}[\min_{e \in R} i_e] \quad = \quad i_R^\alpha + \sum_{k=1}^{\infty} \Pr[2^{k-1} \cdot i_R^\alpha + 1 \leq i_R \leq 2^k \cdot i_R^\alpha] \cdot 2^k \cdot i_R^\alpha$$

$$\leq \quad i_R^\alpha + \sum_{k=1}^{\infty} 2^k \cdot i_R^\alpha \cdot e^{-kQ\alpha} = i_R^\alpha/(1 - 2e^{-Q\alpha})$$

$\square$

**Lemma 9.** *Let $Q := z/\alpha$ for some positive constant $z$. For any request $R$ with covering requirement $\mathrm{K}(R) = 1$,*

$$\mathbb{E}_{\pi \sim \mathrm{P}^A} [\mathrm{AccessCost}(\pi, R)] \leq \left( \frac{4z}{1 - 2e^{-z}} + 1 \right) \cdot \sum_{i=1}^{n} \left( 1 - \sum_{j=1}^{i-1} \sum_{e \in R} A_{ej} \right)_+$$

*Proof.*

$$\mathbb{E}_{\pi \sim \mathrm{P}^A}[\mathrm{AccessCost}(\pi, R)] \quad = \quad \int_0^1 \mathbb{E}_{\pi \sim \mathrm{P}_\alpha^A}[\mathrm{AccessCost}(\pi, R)] \cdot (2\alpha) \, d\alpha$$

$$\leq \quad \int_0^1 2Q \cdot \mathbb{E}[\min_{e \in R} i_e] \cdot (2\alpha) \, d\alpha + \int_0^1 (2\alpha) \, d\alpha \quad (Lemma\ 6)$$

$$= \quad \int_0^1 4z \cdot \mathbb{E}[\min_{e \in R} i_e] d\alpha + 1 \quad (Q = z/\alpha)$$

$$\leq \quad \int_0^1 4z \cdot i_R^\alpha/(1 - 2e^{-z}) \, d\alpha + 1 \quad (Lemma\ 8\ \text{and}\ Q = z/\alpha)$$

$$= \quad \frac{4z}{1 - 2e^{-z}} \int_0^1 i_R^\alpha \, d\alpha + 1 \quad (z = \alpha Q\ \text{is constant})$$

$$\leq \quad \left( \frac{4z}{1 - 2e^{-z}} + 1 \right) \sum_{i=1}^{n} \left( 1 - \sum_{j=1}^{i-1} \sum_{e \in S} A_{ej} \right)_+ \quad (Lemma\ 8)$$

$$= \quad \left( \frac{4z}{1 - 2e^{-z}} + 1 \right) \mathrm{SW}_R(A) \quad (Corollary\ 2)$$

$\square$

Lemma 4 directly follows by setting $z := 1.6783$ in Lemma 9.

We conclude the section with the following corollary that provides with a quadratic-time algorithm for computing the subgradient in case of *Min-Sum Set Cover problem*.

**Corollary 3.** *Let a doubly stochastic matrix $A$ and a request $R$. Let $i^*$ denotes the index at which $\sum_{j=1}^{i-1} A_{ej} \leq 1$ and $\sum_{j=1}^{i-1} A_{ej} > 1$. Let also the $n \times n$ matrix $B$ defined as follows,*

$$B_{ej} = \begin{cases} i^* - j & \text{if } j \leq i^* - 1 \text{ and } e \in R \\ 0 & \text{otherwise} \end{cases}$$

*The matrix $B$ (vectorized) is a subgradient of the $SW_R(\cdot)$ at point $A$.*

## B.3 Proof of Theorem 2

All steps of Algorithm 2 run in polynomial-time. In Step 3 of Algorithm 2, any $(1+\alpha)$-approximation, polynomial-time algorithm for $\min_{R\in[\text{Rem}]^r} \text{AccessCost}(R, A)$ can be used. The first choice that comes in mind is exhaustive search over all the requests of size $r$, resulting in $\Theta(n^r)$ time complexity. Since the latter is not polynomial, we provide a $(1+\alpha)$-approximation algorithm running in polynomial-time in both parameters $n$ and $r$. For clarity of exposition the algorithm used in Step 3 is presented in Section B.5. In the following we focus on proving Theorem 2.

We remark that by Corollary 2 of Section B.2 and Lemma 3 of Section B.1, for any request $R$ with covering requirement $\text{K}(R) = 1$,

$$\sum_{i=1}^{n} \left( 1 - \sum_{j=1}^{i-1} \sum_{e\in R} A_{ej}, 0 \right)_+ \leq (1+\epsilon) \cdot \text{FAC}_R(A) \text{ for any } A \in \text{DS}$$

where $\epsilon$ is the parameter used in Definition 2. As a result, Theorem 2 follows directly by Theorem 5, which is stated below and proved in the next section.

**Theorem 5.** *Let $\pi_A \in [n!]$ denote the permutation of elements produced by Algorithm 2 when the doubly stochastic matrix $A \in \text{DS}$ is given as input. Then for any request $R$ with $|R| \leq r$ and $\text{K}(R) = 1$,*

$$\text{AccessCost}(\pi_A, R) \leq 2(1+\alpha)^2 r \cdot \sum_{i=1}^{n} \left( 1 - \sum_{j=1}^{i-1} \sum_{e\in R} A_{ej} \right)_+ .$$

## B.4 Proof of Theorem 5

Consider a request $R \in [n^r]$ such that

$$(L-1) \cdot r + 1 \leq \text{AccessCost}(\pi_A, R) \leq L \cdot r \tag{5}$$

for some integer $L$. Since $\text{K}(R) = 1$ this means that the first element of $R$ appears between positions $(L-1) \cdot r + 1$ and $L \cdot r$ in permutation $\pi_A$.

To simplify notation we set $\text{Cost}(A, R) := \sum_{i=1}^{n} \left( 1 - \sum_{j=1}^{i-1} \sum_{e\in R} A_{ej} \right)_+$. To prove Theorem 5 we show the following, which can be plugged in (5) and give the result:

$$\text{Cost}(A, R) \geq \frac{L}{2(1+\alpha)^2}.$$

Let $R_\ell$ denote the request of size $r$ composed by the elements lying from position $(\ell-1) \cdot r + 1$ to $\ell \cdot r$ in the produced permutation $\pi_A$. Recall the minimization problem of Step 3. $R_\ell$ is a $(1+\alpha)$ approximately optimal solution for that problem and thus its corresponding cost is at most $(1+\alpha)$ times the corresponding cost of any other same-cardinality subset of the remaining elements. Since in $\pi_A$ all the elements of $R$ lie on the right of position $(L-1) \cdot r$, all elements of $R$ are present at the $L$-th iteration and thus,

$$\text{Cost}(A, R_L) \leq (1+\alpha) \cdot \text{Cost}(A, R)$$

Moreover, by the same reasoning,

$$\text{Cost}(A, R_\ell) \leq (1+\alpha) \cdot \text{Cost}(A, R_L), \text{ for all } \ell = 1, \dots, L.$$

Thus it suffices to show that $\text{Cost}(A, R_L) \geq L/2(1+\alpha)$. The latter is established in Lemma 10, which concludes the section.

**Lemma 10.** *Let $R_1, R_2, \dots, R_L$ be disjoint requests of size $r$ such that for all $\ell = 1, \dots, L$, $\text{Cost}(A, R_\ell) \leq (1+\alpha) \cdot \text{Cost}(A, R_L)$. Then,*

$$\text{Cost}(A, R_L) \geq \frac{L}{2(1+\alpha)}$$

*Proof.* For each request $R_\ell$ we define the quantity $B_{\ell i}$ as follows:

$$B_{\ell i} = \begin{cases} \sum_{e\in R_\ell} A_{ei} & \text{if } \sum_{j=1}^{i} \sum_{e\in R_\ell} A_{ej} < 1 \\ 1 - \sum_{j=1}^{i-1} \sum_{e\in R_\ell} A_{ej} & \text{if } \sum_{j=1}^{i} \sum_{e\in R_\ell} A_{ej} \geq 1 \text{ and } \sum_{j=1}^{i-1} \sum_{e\in R_\ell} A_{ej} < 1 \\ 0 & \text{otherwise} \end{cases}$$

**Observation 1.** *The following 3 equations hold,*

1. $\sum_{i=1}^{n} B_{\ell i} = 1$.

2. $B_{\ell i} \leq \sum_{e \in R_\ell} A_{ei}$.

3. $\mathrm{Cost}(A, R_\ell) = \sum_{i=1}^{n} \left(1 - \sum_{j=1}^{i-1} \sum_{e \in R_\ell} A_{ej}\right)_+ = \sum_{i=1}^{n} \left(1 - \sum_{j=1}^{i-1} B_{\ell j}\right)$

Since $(1 + \alpha) \cdot \mathrm{Cost}(A, R_L) \geq \mathrm{Cost}(A, R_\ell)$ for all $\ell = 1, \ldots, L$,

$$
\begin{aligned}
\mathrm{Cost}(A, R_L) &\geq \frac{1}{1+\alpha} \cdot \frac{1}{L} \sum_{\ell=1}^{L} \mathrm{Cost}(A, R_\ell) = \frac{1}{1+\alpha} \cdot \left[\frac{1}{L} \sum_{\ell=1}^{L} \sum_{i=1}^{n} \left(1 - \sum_{j=1}^{i-1} B_{\ell j}\right)\right] \\
&= \frac{1}{1+\alpha} \cdot \left[n - \frac{1}{L} \sum_{\ell=1}^{L} \sum_{i=1}^{n} \sum_{j=1}^{i-1} B_{\ell j}\right] \\
&= \frac{1}{1+\alpha} \cdot \left[n - \frac{1}{L} \sum_{i=1}^{n} \sum_{j=1}^{i-1} C_j\right] \quad \text{(where } C_j = \sum_{\ell=1}^{L} B_{\ell j}) \\
&= \frac{1}{1+\alpha} \cdot \left[n - \frac{1}{L} \sum_{i=1}^{n} (n-i) \cdot C_i\right] = \frac{1}{1+\alpha} \cdot \left[n - \frac{n}{L} \sum_{i=1}^{n} C_i + \frac{1}{L} \sum_{i=1}^{n} i \cdot C_i\right]
\end{aligned}
$$

Observe that $\sum_{i=1}^{n} C_i = \sum_{i=1}^{n} \sum_{\ell=1}^{L} B_{\ell i} = \sum_{\ell=1}^{L} \sum_{i=1}^{n} B_{\ell i} = L$, where in the last equality we used $\sum_{i=1}^{n} B_{\ell i} = 1$. Thus we get that

$$
\mathrm{Cost}(A, R_L) \geq \frac{1}{1+\alpha} \left[\frac{1}{L} \sum_{i=1}^{n} i \cdot C_i\right]
$$

To this end, to conclude the result, one can prove that $\sum_{i=1}^{n} i \cdot C_i \geq L^2/2$ using that $\sum_{i=1}^{n} C_i = L$ and $C_i \leq 1$. $C_i \leq 1$ follows by the *disjoint property* of the requests $R_1, \ldots, R_L$. More precisely,

$$
\begin{aligned}
C_i &= \sum_{\ell=1}^{L} B_{\ell i} \leq \sum_{\ell=1}^{L} \sum_{r \in R_\ell} A_{ri} \\
&\leq \sum_{e=1}^{n} A_{ei} = 1
\end{aligned}
$$

where the first inequality follows from Observation 1 and the last inequality by $R_1, \ldots, R_L$ not sharing any element. □

## B.5 Implementing Step 3 of Algorithm 2 in Polynomial-Time

In this section we present a polynomial time algorithm implementing Step 3 of Algorithm 2. More precisely, we present a *Fully Polynomial-Time Approximation Scheme (FTPAS)* for the combinatorial optimization problem defined below, in Problem 1.

**Problem 1.** *Given an $n \times n$ doubly stochastic matrix $A$ and a set of elements $Rem \subseteq \{1, \ldots, n\}$. Select the $r$ elements of $Rem$ ($R^* \subseteq Rem$ with $R^* = r$) minimizing,*

$$
\sum_{i=1}^{n} \left(1 - \sum_{j=1}^{i-1} \sum_{e \in R^*} A_{ej}\right)_+ .
$$

In fact we present a $(1 + \alpha)$-approximation algorithm for a slightly more general problem, Problem 2.

**Problem 2.** *Given a set of $m$ vectors $B_1, \ldots, B_m$, of size $n$ such that,*

$$0 = B_{e1} \leq B_{e2} \leq \ldots \leq B_{en} = 1, \text{ for each } e = 1, \ldots, m$$

*Select the $r$ vectors ($R^* \subseteq [m]$ with $R^* = r$) minimizing*

$$\sum_{i=1}^{n} \left( 1 - \sum_{e \in R^*} B_{ei} \right)_+$$

Setting $B_{ei} = \sum_{j=1}^{i-1} A_{ej}$, one can get Problem 1 as a special case of Problem 2.

**Theorem 6.** *There exists a $(1+\alpha)$-approximation algorithm for Problem 2 that runs in $\Theta(n^4 r^3 / \alpha^2)$ steps.*

The $(1+\alpha)$-approximation algorithm of Problem 2 heavily relies on solving the Integer Linear Program defined in Problem 3.

**Problem 3.** *Given a set of $m$ triples of integers $(w_e, c_e, d_e)$ such that $c_e, d_e \geq 0$ for each $e \in \{1, m\}$ and two positive integers $C, D$,*

$$
\begin{aligned}
\text{minimize} \quad & \sum_{e=1}^{m} w_e x_e \\
\text{subject to} \quad & \sum_{e=1}^{m} c_e x_e \geq C \\
& \sum_{e=1}^{m} d_e x_e \leq D \\
& \sum_{e=1}^{m} x_e = r \\
& x_e \in \{0, 1\} \quad e = 1, \ldots, m
\end{aligned}
$$

**Lemma 11.** *Problem 3 can be solved in $\Theta(n \cdot C \cdot D \cdot r)$ steps via Dynamic Programming.*

*Proof.* Let $\mathrm{DP}(n, r, C, D)$ denotes the value of the optimal solution. Then

$$\mathrm{DP}(n, r, C, D) = \min\left( \mathrm{DP}(n-1, r-1, C - x_n, D - d_n), \mathrm{DP}(n-1, r, C, D) \right)$$

$\square$

In the rest of the section, we present the $(1+\alpha)$-approximation algorithm for Problem 2 as stated in Lemma 6 using the algorithmic primitive of Lemma 11.

We first assume the entries of the input vectors are multiples of small constant $\alpha \ll 1$, $B_{ei} = k_{ei} \cdot \alpha$ for some integer $k_{ei}$. Under this assumption we can use the algorithm (stated in Lemma 11) for Problem 3 to find the *exact* optimal solution of Problem 2 in $\Theta(n^2 r / \alpha^2)$ steps.

More precisely, for a fixed index $k$, let $\mathrm{OPT}_k$ denotes the optimal solution among the set of vectors of size $r$ that additionally satisfy,

$$\sum_{e \in R} B_{e(k-1)} < 1 \text{ and } \sum_{e \in R} B_{ek} \geq 1 \tag{6}$$

It is immediate that $\mathrm{OPT} = \arg\min_{1 \leq k \leq n} \mathrm{OPT}_k$ and thus the problem of computing $\mathrm{OPT}$ reduces into computing $\mathrm{OPT}_k$ for each index $k$. We can efficiently compute $\mathrm{OPT}_k$ for each index $k$ by solving an appropriate instance of Problem 3. To do so, observe that for any set of vectors $R$ satisfying the constraints of Equation (6) for the index $k$,

$$\sum_{i=1}^{n} \left( 1 - \sum_{e \in R} B_{ei} \right)_+ = \sum_{i=1}^{k-1} \left( 1 - \sum_{e \in R} B_{ei} \right) = \sum_{e \in R} \underbrace{\sum_{i=1}^{k-1} \left( \frac{1}{r} - B_{ei} \right)}_{w_e}$$

where the first equality comes from the fact that $B_{e1} \leq \ldots \leq B_{en}$. It is not hard to see that $\mathrm{OPT}_k$ can be computed via solving the instance of Problem 3 with triples

$\left(w_e = \sum_{i=1}^{k-1}\left(\frac{1}{r} - B_{ei}\right), c_e = B_{e(k-1)}, d_e = B_{ek}\right)$ for each $e = 1\ldots, m$, $D = 1$ and $C = 1$. Moreover by Lemma 11 this is done in $\Theta(nr/\alpha^2)$ steps. Thus the overall time complexity in order to compute the optimal solution of Problem 2 (in case the entries $B_{ei}$ are multiplies of $\alpha$) is $\Theta(n^2 r/\alpha^2)$.

We now remove the assumption that the entries $B_{ei}$ are multiples of $\alpha$ via relaxing the optimality guarantees by a factor of $(1 + \alpha)$. We first construct a new set of vector with entries *rounded* to the closest multiple of $\alpha$, $\hat{B}_{ei} = \lfloor B_{ei}/\alpha \rfloor \cdot \alpha$ and solve the problem as if the entries where $\hat{B}_{ei}$ in $\Theta(n^2 r/\alpha^2)$ steps. The quality of the produced solution, call it Sol can be bounded as follows

$$
\begin{aligned}
\sum_{i=1}^{n}\left(1 - \sum_{e \in \text{Sol}} B_{ei}\right)_+ &\leq \sum_{i=1}^{n}\left(1 - \sum_{e \in \text{Sol}} \hat{B}_{ei}\right)_+ \\
&\leq \sum_{i=1}^{n}\left(1 - \sum_{e \in \text{Sol}} \hat{B}_{ei}\right)_+ \\
&\leq \sum_{i=1}^{n}\left(1 - \sum_{e \in \text{OPT}} (B_{ei} - \alpha)\right)_+ \\
&\leq \sum_{i=1}^{n}\left(1 - \sum_{e \in \text{OPT}} B_{ei}\right)_+ + nr \cdot \alpha
\end{aligned}
$$

Setting $\alpha := \alpha'/nr$, we get that

$$
\sum_{i=1}^{n}\left(1 - \sum_{e \in \text{Sol}} B_{ei}\right)_+ \leq \sum_{i=1}^{n}\left(1 - \sum_{e \in \text{OPT}} B_{ei}\right)_+ + \alpha' \leq (1 + \alpha') \sum_{i=1}^{n}\left(1 - \sum_{e \in \text{OPT}} B_{ei}\right)_+
$$

since $B_{e1} = 0$ for all $e$. Thus, the overall time needed to produce a $(1 + \alpha')$-approximate solution is $\Theta(n^4 r^3/(\alpha')^2)$, proving the result.

### B.6 A simple heuristic for Problem 1

In this section we present a simple heuristic for Problem 1 that can be a good alternative of the algorithm elaborated in Section B.5. We remark that Algorithm 5 may provide highly sub-optimal solutions in the worst case however our experiments suggest that it works well enough in practice. As explained in Section 5, in our experimental evaluations we use this heuristic to implement Step 3 of Algorithm 2. This was done since this heuristic is easier and faster to implement.

---

**Algorithm 5** A simple heuristic for Problem 1

---

**Input:** A doubly stochastic matrix $A \in \text{DS}$.
**Output:** A set $R$ (of $r$ elements) approximating Problem 1
1: $R = \varnothing$
2: Target $= \underbrace{(1, \ldots, 1)}_{n}$
3: **for** $\ell = 1$ to $r$ **do**
4:     $e_\ell \leftarrow \arg\min_{e \in \{1,\ldots,n\}/R}\left(\sum_{j=1}^{n}\max(\text{Target}[j] - \sum_{s=1}^{j-1} A_{ej}, 0)\right)$
5:     $R \leftarrow R \cup \{e_\ell\}$
6:     Target $\leftarrow (\text{Target} - (A_{e1}, \ldots, A_{en}))_+$
7: **end for**
8: Output the set of elements $R$.

---