[Reviews · NeurIPS 2020]

Review 1

Summary and Contributions: In this paper, the authors consider the Generalized Min-Sum-Set-Cover (GMSSC) problem under an online learning setting. Due to the NP-hardness of the problem, the current non-regret methods are computationally inefficient. The main contributions are 1. The authors prove that there exists an algorithm for online GMSSC in polynomial-time meanwhile obtains a low regret. 2. A key idea of the algorithm is to relax the original problem (in the space of permutation) into the space of doubly stochastic matrices. Thus a polynomial-time algorithm exists.

Strengths: 1. The authors consider the GMSSC problem under an online learning setting, which is new and theoretically interesting. 2. The theoretical contributions to this new problem are solid. 3. The proposed method can run in polynomial-time with a low regret guarantee. The idea of relaxing the original problem into the space of doubly stochastic matrices is technically interesting.

Weaknesses: 1. The designed algorithm may not be empirically efficient. At each iteration, the key step of Line 5 of Algorithm 1 needs to solve a convex problem based on the ellipsoid method, which is impractical especially when n goes large. The problem of online learning setting is motivated from the real-world application where each iteration (a small response time) needs to be very efficient. For example, in the application of web search, the parameter n is usually very large or maybe larger than T in some cases? 2. It would be more interesting if there is any simulation study and check how these regrets decrease as the time T goes large.

Correctness: Claims and methods are correct. The empirical methodology is also correct.

Clarity: The presentation is clear. The paper is well written.

Relation to Prior Work: Yes. The authors clearly discussed the related work in the Introduction section.

Reproducibility: No

Additional Feedback: In general, the authors propose new methods for an important problem under the online learning setting. The theoretical contributions are solid. The presentation is clear. However, my only concern is from a practical point view. I would argue that the designed algorithm may not be empirically efficient. At each iteration, the key step of Line 5 of Algorithm 1 needs to solve a convex problem based on the ellipsoid method, which is impractical especially when n goes large. Q1. I was wondering if it is possible to reduce the approximation complexity \mathcal{O}(n^5.5)} due to the norm of approximated subgradients. Could the authors discuss a little bit more about this? One typo: Typos: Decent -> Descent


Review 2

Summary and Contributions: The paper presents new algorithms utilizing online projected gradient descent(OPGD) that solves generalized min-set sum cover (GMSSC) problem in polynomial time with approximately 28-regret and solves min-set sum cover (MSCC) problem in polynomial time with approximately 11-regret. Compared with previous result that focuses on offline problems, this paper focuses on online version and shows even in online set up, low-regret solution can still be obtained in polynomial time. --- Update based on the Author Response The author addressed some of my concerns, but my overall assessment of the paper remains the same.

Strengths: The novelty lies on the use of OPGD and how this can be translated into a permutation of items for online requests which yields better solution (in the sense of lower regret) than previous results. The proof of the theoretical result is rigorous as well, ranging from the main result to showing some key steps can be computed in polynomial time.

Weaknesses: Some of the results in the paper can be viewed as minor improvement/modification over the result from ``Preemptive and non-preemptive generalized min sum set cover'' written by Im S, Sviridenko M, Zwaan R (2013) and ``A note on the generalized min-sum set cover problem'' written by Martin Skutella and David P. Williamson(2011). From this perspective, there is indeed novelty, but the novelty is not significant enough to be viewed as strong submission.

Correctness: Yes.

Clarity: There is much room for improvement in writing. Several typos can be found in the proof, for example in the proof of Lemma 7, there are several summations missing; in the proof of Lemma 8, there is e ∈ S which should be e ∈ R, in the proof of Lemma 9, the last inequality shall follow Lemma 7 instead of Lemma 8. There are also typos in the statement of Lemma 5 and in other places in the proof. For audience not familiar with this field, this will lead to confusion in reading. Besides, more explanation can be given instead of purely stating ”as a result” or ”from manipulation of definition”.

Relation to Prior Work: Yes, the paper gives the result of previous literature and show how it differs from previous ones.

Reproducibility: Yes

Additional Feedback: The writing can be improved. Besides, it might be a good idea to discuss the coefficients (5.03 in algorithm 3 and 1.67 in Algorithm 4). Also, the authors obtained the coefficient in algorithm 4 through optimizing4 z/(1−2e^{-z}). However, for different K(R), there might exist a different optimal coefficient. A natural question is whether a general principle exists. In addition, I will recommend cleaning up the writing not only to correct for the typos but also to explain the proof in more details.


Review 3

Summary and Contributions: The authors consider online preference aggregation - wherein a set of preferred items $R_1, ..R_t$ is given in each round, along with the demand for any $k_t$ items within the set $R_t$. The learner maintains a ranking $\pi_t$ of items. The objective is to minimize the rank at which the $k_t$th item from $R_t$ appears in the ranking $\pi_t$. This problem is the online version of the generalized min sum set cover (GMSSC) problem, which as a special case subsumes the well-studied min sum set cover (MSSC) problem. The MSSC is NP-hard and inapproximable beyond a factor of 4. To develop a no-regret online learning algorithm, the authors learn over doubly stochastic matrices (instead of the permutahedron) using the MWU algorithm and consider a relaxation for the GMSSC that approximates the actual objective with a constant factor, thereby obtaining a polynomial time algorithm.

Strengths: The GMSSC problem is interesting in that it goes beyond learning over rankings, and applies broadly to many preference aggregation settings. The minimum linear ordering problem is challenging, and MSSC is a special case of this. Learning over doubly stochastic matrices instead of the convex hull of permutations is quite standard, however, the change of the objective to get a convex relaxation to the GMSSC objective is interesting. -------- ------- EDIT: I thank the authors for their response.

Weaknesses: 1. In the definition of the fractional access cost, the authors assume a fixed accuracy parameter, \epsilon, and ignore dependence of FAC_R(A) on \epsilon. What is the dependence on \epsilon? 2. Writing in unclear. What is the fractional access cost linear program? Why is it exponential in number of constraints? Is this the only possible formulation? Where does the n^4/\epsilon term come from? Why is there a 5.03/\alpha in the conversion of doubly stochastic matrices to probability distribution over permutations? Why doesn't any conversion that respects the marginals suffice? I suspect claim 1 in the appendix is crucial to understanding why $n^4/\epsilon$ is the right scaling. 3. What do the algorithms presented here give for the special case of $|R_t|=1$, i.e. the setting in Seuhiro et a. [39]? 4. Can the relaxation of the GMSSC objective over the doubly stochastic matrices improve some known approximation bounds for the minimum linear ordering problem or its special cases, i.e. what is the implication on offline optimization?

Correctness: The key claims of the paper look correct indeed, however I am not completely sure about the constants.

Clarity: The linear programs, discussions, notation, rationale for choice of parameters - could have been written much more transparently and clearly.

Relation to Prior Work: Yes, related work is thorough.

Reproducibility: Yes

Additional Feedback: Some more explanations of why MSSC results would not improve on the constants would be good, they are currently mentioned fleetingly in footnotes and do not have enough detail.


Review 4

Summary and Contributions: The paper studies a variant of online ranking problem. In the offline settings preferred items belong to different groups, and one need to generate a sequence of items so that qualitatively speaking for each group R_t of items at least $k_t$ elements appear high in the list. More formally they formulate the problem as an online version of the known "Generalized Min-Sum Set Cover" (GMSSC) task: In this problem, given a set $U = {1,...,n}$ of $n$ items, in each step 1. The learner selects a permutation $\pi_t$ items in $U$ 2. The adversary selects a request $R_t \subset U$ with demand $k_t$. 3. The learner incurs a cost equal to the access cost of $R_t$ with respect to $\pi_t$ which is defined as smallest index $i$, so that at least $k_t$ elements of $\R_t$ appear in the first $i$ elements of $\pi$. The goal is to minimize the multiplicative regret, that is the ratio of the total cost to the cost of selecting a fixed optimal permutation $\pi*$ in all steps. a strategy has $alpha$-regret, if its cost is at most alpha times the optimum. The problem has been mostly studied in the offline setting before, and there exists an $O(1)$-approximation algorithm for that. The only previous result for the online settings is given by [1], which shows a $1$-regret (just an additive regret) strategy exists, however, that strategy can not be implemented in a polynomial time. It is largely because the the exponentially large action space of the problem (n! permutations). In general such a result can not be obtained as even in the offline version it is NP-hard to get a better than $4$-approximation algorithm. It is largely because the the exponentially large action space of the problem (n! permutations). They first map the permutation space into the space of doubly stochastic matrices, thus offering a continuous relaxation of the problem. Then, they can achieve a $1$-regret algorithm for the continuous problem using fairly standard techniques. The most technical part of the paper is to project the continuous solution back into the permutation space. They give O(1)-approximation rounding scheme which leads to a O(1)-regret algorithm for the original problem. 1. Dimitris Fotakis, Loukas Kavouras, Grigorios Koumoutsos, Stratis Skoulakis, and Manolis351 Vardas. The online min-sum set cover problem. ICALP 2020

Strengths: I think the ideas are very theoretically interesting; I expect that the proposed relaxation can be exploited for other variants of the ranking problem, and more generally other online algorithms where the actions space is the permutation space.

Weaknesses: My main critisim to the paper is the lack of empirical results. The problem seems very practical especailly in recommendation systems, but the paper does not provide any empirical results for their method. The description of the algorithm is clear and the enough details are provided for reader to implement their method, though. Edit: This is partially addressed in the authors' feedback.

Correctness: Theoretically I verified the proofs. No empirical results is provided.

Clarity: The results and proofs are well-explained.

Relation to Prior Work: yes.

Reproducibility: Yes

Additional Feedback:

[Author Response · NeurIPS 2020]

*We thank all the reviewers for their feedback, their apt comments and questions.* We will follow the comments of
Reviewer 2 and 3 to ameliorate the write up and the suggestions of Reviewer 1 and 4 for experimental results.

**Reviewer 1:** The reviewer's comment on the practical efficiency of the ellipsoid method is a valid point. Unfortunately
circumventing the ellipsoid method for the GMSSC seems a very hard task even for offline algorithms. The reason is
that (to the best of our knowledge) there is no linear relaxation of GMSSC with polynomial description and constant
integrality gap. However, for the important special case of *Min-Sum Set Cover* the subgradient can be computed via
a quadratic-time algorithm. For the sake of simpler exposition we did not present it in the original draft but we will
definitely add it. We have efficiently implemented in C++ both OPDG in doubly stochastic matrices and randomized
/deterministic roundings. Our first experimental results (also requested by Rev. 4) reveal that the regret becomes very
small quickly given random requests, while the deterministic rounding performs extremely better than its theoretical
guarantees and even outperforms the randomized rounding. Finally concerning the reviewer's question on reducing the
$n^{5.5}$ of the additive term, we conjecture that this is indeed possible (may be with some increase in the regret bound).
Actually our experimental findings indicate that the right additive term is $n^2$. However, our current analysis is tight and
novel ideas would be needed.

**Reviewer 2:** The reviewer's point on our novelty concerns the randomized rounding scheme (Algorithm 3) that comes
from [36] and possibly Algorithm 4 that is based on the previous one. We just want to mention that a key contribution
of the paper is combining a modification of the configuration LP in [30] (that differs form the LP in [36]) with the
rounding scheme of [36] to obtain constant regret algorithms. So, we bring together the right ideas and techniques from
previous work, after properly adapting them. Moreover, our deterministic rounding is novel. We find the reviewers
question on optimizing the rounding in [36] for other special case of $K(R)$ very interesting even for the offline case.
Addressing it is, however, beyond our current scope, since our analysis concluding in Lemma 9 crucially uses the fact
that $K(R) = 1$.

**Reviewer 3: In the definition of the fractional access cost, the authors assume a fixed accuracy parameter, $\epsilon$,**
**and ignore dependence of $\mathrm{FAC}_R(A)$ on $\epsilon$. What is the dependence on $\epsilon$?** The fractional access cost, $\mathrm{FAC}_R(A)$,
can be defined for any value $\epsilon > 0$. The smaller the choice of $\epsilon$ is the better the regret bounds are $(2(1 + \epsilon)r, 28(1 + \epsilon)$
and $11.713(1 + \epsilon))$ and the greater the additive term becomes $O\left(\frac{n^{5.5}}{\epsilon\sqrt{t}}\right)$. **What is the fractional access cost linear**
**program? Why is it exponential in number of constraints? Is this the only possible formulation?** It is the linear
program in Definition 2 which has exponential size since the number of different configurations is exponential in the
number of elements. As already mentioned, we are not aware of any linear relaxation for GMSSC with polynomial
description and constant integrality gap. **Where does the $n^4/\epsilon$ term come from?** The term $n^4/\epsilon$ plays two important
roles: $i)$ it ensures an upper bound on the norm of the subgradients (which is necessary for the OPGD to run), and
$ii)$ it associates the fractional access cost with the value of the (different) linear relaxation used in [6]. **Why is there**
**a $5.03/\alpha$ in the conversion of doubly stochastic matrices to probability distribution over permutations?** The
selection of $5.03$ is so as to minimize a parametric upper bound (see [36]). **Why doesn't any conversion that respects**
**the marginals suffice?** Unfortunately, such intuitive rounding schemes do not always work. In fact there are cases
where a probability distribution respecting the marginals of the doubly stochastic matrix leads to arbitrarily higher
expected accessing cost than the fractional access cost of the doubly stochastic matrix. For example, for the matrix
$A$ below, selecting with probability $1/2$ the permutation $\{1, 2, 3, 4, \ldots, n - 2, n - 1, n\}$ and with probability $1/2$
the permutation $\{n - 1, n, 3, 4, \ldots, n - 2, 1, 2\}$, respects the marginals. Let the request $R = \{1, 2\}$ with covering
requirements $K(R) = 1$. Its expected access cost is $\Theta(n)$ while its *fractional access cost* under $A$ is $\Theta(1)$. **The**
**algorithms presented here give for the special case of $|R_t| = 1$?** The exact same bounds since they do not take
into account the cardinality of the sets. However it is easy to design 2-regret algorithms for the special case where
$|R_t| = 1$. **Can the relaxation of the GMSSC objective over the doubly stochastic matrices improve some known**
**approximation bounds for the minimum linear ordering problem or its special cases?** The relaxation of the
GMSSC cannot help in the more general linear ordering problem. The reason is that the dual cannot be solved in
polynomial time, since the separation oracle crucially depends on the specific structure of GMSSC.

$$A = \begin{pmatrix} 1/2 & 0 & 0 & \cdots & 0 & 1/2 & 0 \\ 0 & 1/2 & 0 & \cdots & 0 & 0 & 1/2 \\ 0 & 0 & 1 & \cdots & 0 & 0 & 0 \\ \vdots & \vdots & \vdots & \ddots & \vdots & \vdots & \vdots \\ 0 & 0 & 0 & \cdots & 1 & 0 & 0 \\ 1/2 & 0 & 0 & \cdots & 0 & 1/2 & 0 \\ 0 & 1/2 & 0 & \cdots & 0 & 0 & 1/2 \end{pmatrix}$$

[Meta-Review · NeurIPS 2020]

The reviewers have some concerns that the rebuttal didn't fully address but they all like the paper and agree that it should be accepted at NeurIPS. Congratulations!